# Condenser-Type Heat Exchanger for Compost Heat Recovery Systems

**Jaroslav Bajko** *, **Jan Fišer and Miroslav Jícha**

Department of Thermodynamics and Environmental Engineering, Faculty of Mechanical Engineering, Brno University of Technology, 616 69 Brno, Czech Republic; fiser@fme.vutbr.cz (J.F.); jicha@fme.vutbr.cz (M.J.)
* Correspondence: bajko@fme.vutbr.cz; Tel.: +420-54-114-3242

**Abstract:** The majority of heat released during composting is contained in latent form of water vapour. To improve the rate of heat recovery, a simple heat exchanger based on condensation of compost vapours was designed. A prototype of this condenser-type heat exchanger was built and tested as a part of pilot-scale compost heat recovery system. Passively aerated static pile (modified Jean Pain mound) with enhanced aeration using vertical channels was chosen for this composting experiment. Insulation of the compost mound and adjacent hoop house further improved the efficiency of the heat recovery and utilization.

**Keywords:** composting; heat recovery; condenser-type heat exchanger; sustainable energy; biomass utilization

---

## 1. Introduction

Composting is well known as a complex aerobic process governed by microorganisms with its inherent temperature dynamics divided into three main phases: psychrophilic, mesophilic and thermophilic. The environment with the highest temperatures is created in the thermophilic phase that occurs immediately after initial heat-up provided that suitable compost mixture [1], its physical and chemical parameters, composting principles [2] as well as proper composting facility construction [3] and management [4] were respected. Compost Heat Recovery Systems (CHRSs) [5] are being reviewed [6], investigated and modeled [7] as well as developed in order to utilize the heat produced by bacterial metabolism. This biologically produced heat can be considered as low-temperature heat source [8] with the potential of becoming another method for sustainable energy [9] production from biomass [10] and a novel waste management strategy [11]. This heat would be otherwise treated as waste heat and dissipated into the environment without productive use. Applications of this heat source are in green and hoop houses, preparation of hot water [12] and heating of buildings [13].

At the pilot-scale or practical level, experiments with the Jean Pain-style composting systems [12] are being conducted and modified (under names such as Biomeiler [14], thermocompost, bioreactor or composting reactor). However, the evidence and measured data of their life cycle, performance and final compost quality are scarce in scientific literature. For the purpose of filling this gap, experiments have been performed [15–18] with medium-sized compost piles (volume 15–20 m$^3$) where the whole process was observed, physical/chemical parameters were measured and potential of this renewable heat source and biomass utilization strategy was estimated.

In this research, we focus on heat capture and its transport mechanisms within the passively aerated static pile with the aim of maximum heat recovery without adversely affecting the composting process. The heat exchange system, especially the type of the heat exchanger, determines the way the heat is recovered. Careful design needs to be applied in order to tap into the most abundant heat source which naturally occurs during composting, i.e., the water vapour [5].

---

Heat exchangers used in Jean Pain style bioreactors are usually formed from plastic pipes that are gradually laid in layers and in a certain pattern as the compost mound is built [19]. This approach allows heat removal from the whole volume of the pile by conduction as a prevailing heat transport mechanism. Limitations of this system are the susceptibility to leakage and potential damage by heavy weight or manipulation [12]. Moreover, the system of installation/dismantling of the heat exchanger is dependent on the pile construction/deconstruction and it can be labour intensive and impractical on larger scales (when machinery is used). Importantly, excess heat removal from internal parts of the pile can affect heat generation [19] and potentially the microbial activity itself [20].

The goals of this study are therefore the introduction of a new type of heat exchanger based on condensation of water vapour, providing evidence, operational data and analysis of the performance of the heat exchanger prototype under real conditions (in a medium-sized compost pile) as well as sharing and discussing observations and drawing conclusions for the further development of compost heat recovery technique.

*Impact of Heat Removal on Its Source*

Since temperature determines which bacteria (from thermophilic, mesophilic or psychrophilic families) predominates in aerobic degradation processes, direct heat removal from internal parts of the compost mound via heat exchanger can therefore affect their activity. Potentially, too much heat removed from the mound can harness the whole composting process as some bacteria become inactive or dormant.

Experiments with continuous temperature measurements inside the compost mound provided an observation of the impact of heat removal on heat sources, [17]. Figure 1 shows temperature profiles inside the mound (left) and the position of measuring poles with sensors and schematic drawing of spiral heat exchanger (right) buried inside the compost mound. Inlet and outlet from the hydronic heat exchanger and ambient temperature sensors are depicted as well as direction of water circulation.

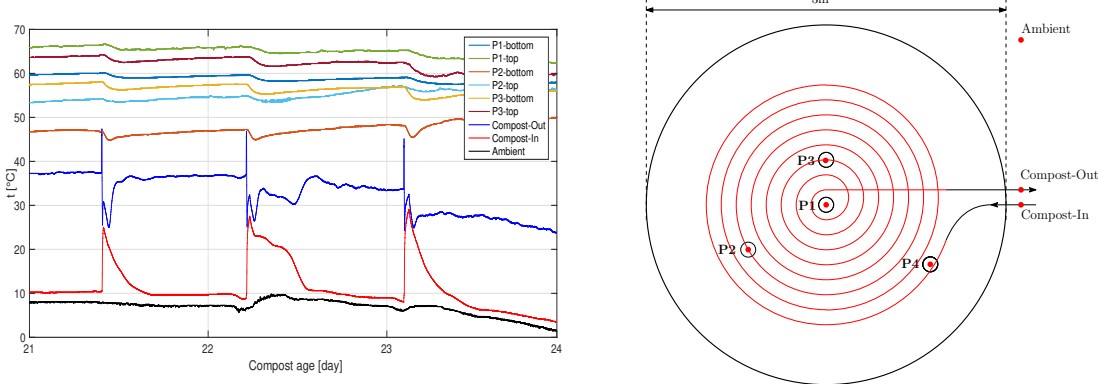

**Figure 1.** (**Left**) Temperature profiles at various points inside the mound; inlet and outlet water; ambient temperature. (**Right**) Position of measuring poles (P1–P4) with sensors and schematic drawing of the spiral heat exchanger.

The graph in Figure 1 recorded three time lapses when the circulation pump was on and heat had been actively removed from the mound. Power output for each 15, 30 and 45-min run was estimated as $P = 1914; 1554$ and $1379$ W, respectively. Even if the heat recovery system was on only for a short period of time, a slight drop in the internal temperature field is noticeable. Moreover, previous steady state (balance between heat production and losses) had not been fully recovered in this experiment and the internal temperatures continued to decrease.

Note that the standalone cylidrical pile, 3 m in diameter, 2 m high, without any insulation, has significant heat losses due to its high surface area/volume ratio 1.83 $m^2/m^3$. Larger piles seem to be less prone to adverse effects of heat removal, although cold inlet water is circulated in the heat exchanger [12,19].

## 2. Materials and Methods

*2.1. Design and Manufacturing of the Condenser-Type Heat Exchanger (CHE)*

The majority of heat in gaseous streams rising from compost piles (either naturally or by forced aeration) is bounded in the liquid–vapour phase change [5]. In order to capture this latent heat source, condensation needs to occur and released heat can be transferred via a suitable heat-carrying medium. Condensed water solutions should be processed or re-used as an active composting process requires maintenence of moisture content 40–60% by weight. Aeration and/or oxygen diffusion, as another key element for composting, should not be compromised by the heat exchanger. Moreover, like the decomposition of the biomass progresses, the heat exchanger needs to cope with the mound deformations and shape changes caused by compaction of the material.

From the previous findings and requirements, the design features of a new heat exchanger can be identified as follows:

- It minimizes or removes the adverse effects on microbial activity caused by direct heat capture and transfer from the pile.
- It allows condensation of water vapour from moist air on its surface.
- Condensed water is re-used for keeping optimal moisture levels inside the pile.
- Aeration of the pile is not compromised by the heat exchanger.
- Installation and dismantling of the heat exchanger is not dependent on the construction of the pile itself.
- It reflects the compaction of the pile in time.

The scale and type of the composting system is also very important in designing the CHRS. For the purpose of this study, let us assume pilot scale composting in static, standalone and passively aerated pile—see Section 2.2 for detailed information on construction.

The proposed heat exchanger should be installed on top of the mound after the main compost body is built. It should utilize the tendency of natural convection to push the hot and moist air into the upper parts of the pile in the thermophilic phase [16]. In passively aerated piles with heat recovery, it is therefore advisable to enhance and intensify the chimney effect and take it into account while the pile is built—see Section 2.4.

Figure 2 shows a schematic drawing of a heat exchanger designed for low density polyethylene (LDPE) pipe (outer diameter 25 mm, wall thickness 3.5 mm) that has been shaped into a spiral form. Total length of the pipe in the heat exchange region is 61.2 m, which corresponds to surface area 4.8 m$^2$. Overall volume of heat-exchange medium inside the pipe is 15.6 L.

Condensation of moist air is allowed on the surface of the heat exchanger since a cavity is created underneath the piping (cf. Figure 3) using a wooden lath structure that serves also for fastening the pipe into the spiral shape. The first prototype of the heat exchanger built from recycled sections of LDPE pipe is shown in Figure 4. The yellow tube indicates the center of the pile and serves as one of the main aeration channels—see Section 2.4.

As water vapours condense on the pipe surface, droplets are formed and drip back to the body of the compost pile. To secure this water retention, the heat exchanger is covered with a layer of porous materials (woodchips, straw and biochar) of about 20 cm thick. Another 30 cm thick layer of matured compost is then laid on top as a biofilter cover layer—see Section 2.3.

Since the heat removal is applied from the top part of the pile, composting processes (and bacterial activity) should be directly affected only in a small volume of the pile compared to hydronic exchangers that are spread across the whole volume of composted material. The space between every semi-circle (see Figure 5) of the pipe allows penetration of gases and the aeration of the biomass should not be compromised. This condenser-type heat exchanger should therefore fulfill all the requirements mentioned in the beginning of this section.

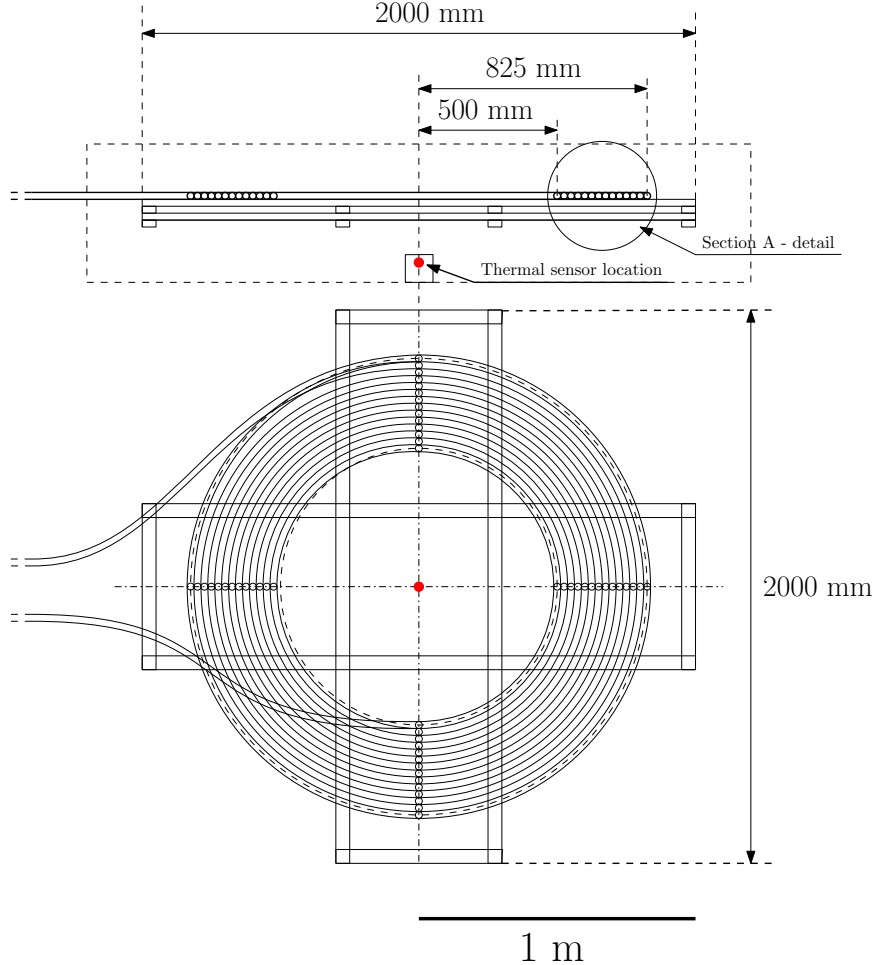

**Figure 2.** A condenser-type Heat Exchanger mounted on a wooden structure allowing airflow underneath the pipe spiral.

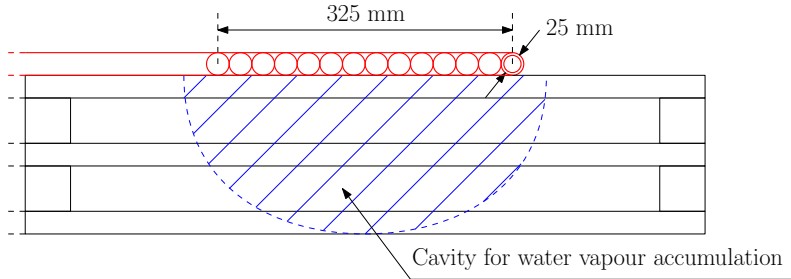

**Figure 3.** Section A—Detail. Underneath the heat exchanger, the cavity is formed for water vapour accumulation.

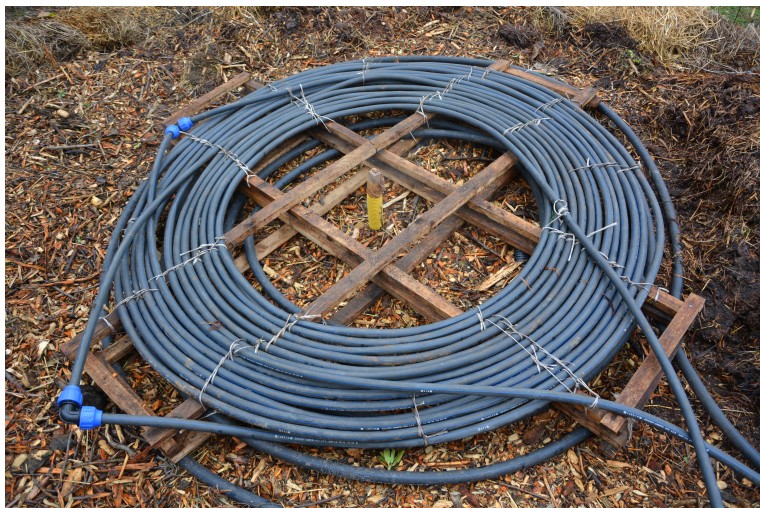

**Figure 4.** Prototype of the heat exchanger: L-shaped fittings connect three shorter sections of the pipe forming a total length of 61.2 m. Wider LDPE pipe visible underneath the exchanger is a potential second heat exchange loop—it had not been used in this experiment.

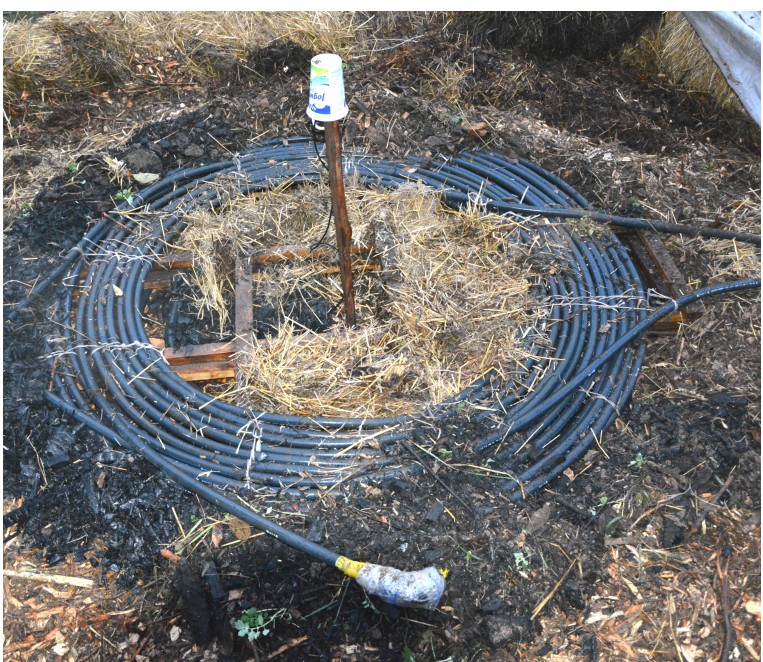

**Figure 5.** The heat exchanger is gradually covered with porous materials. Note that the cavity is formed underneath the piping, allowing condensation to happen. The temperature sensor for a compost core is installed inside the yellow tube while still uncovered. The wire is hidden under the white cup.

## 2.2. Design of the Composting Bioreactor

The compost pile for this experiment is designed as a standalone passively aerated pile [4] with a large D shaped semi-circle ground plan as shown in Figure 6. The pile height is 2 m and the semi-circle diameter of 3 m results in the overall volume 16.07 m$^3$ of composted biomass.

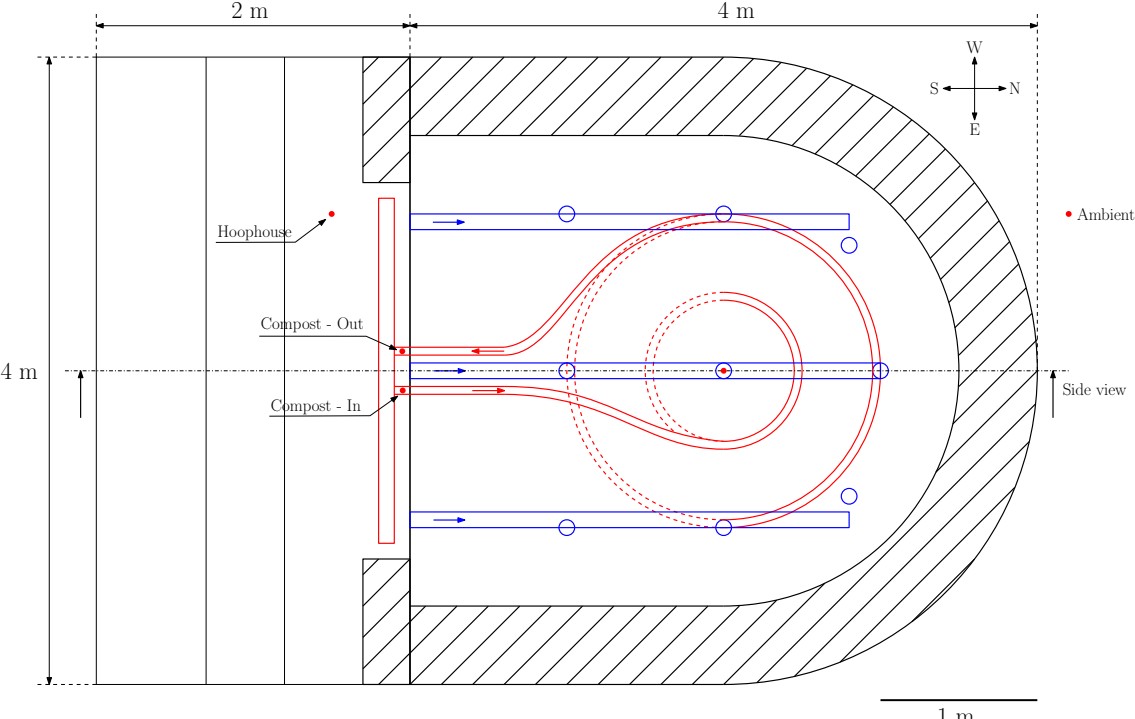

**Figure 6.** Design of the composting bioreactor: Red—schematic drawing of the CHRS; Blue—aeration system; Shaded region—straw bale thermal insulation.

Surface area-to-volume ratio is 1.89 m$^2$/m$^3$, assuming only sides and a top of the pile, i.e., base contact area is excluded. Shaded areas in Figures 6 and 7 represent thermal insulation layers made of compact bales of straw $100 \times 50 \times 30$ cm in size. The red color schematically depicts the CHRS, which includes the heat exchange and utilization zone. Red arrows then indicate the flow direction and red dots the position of temperature sensors. The blue color indicates the aeration channels (vertical and horizontal) and the direction of airflow.

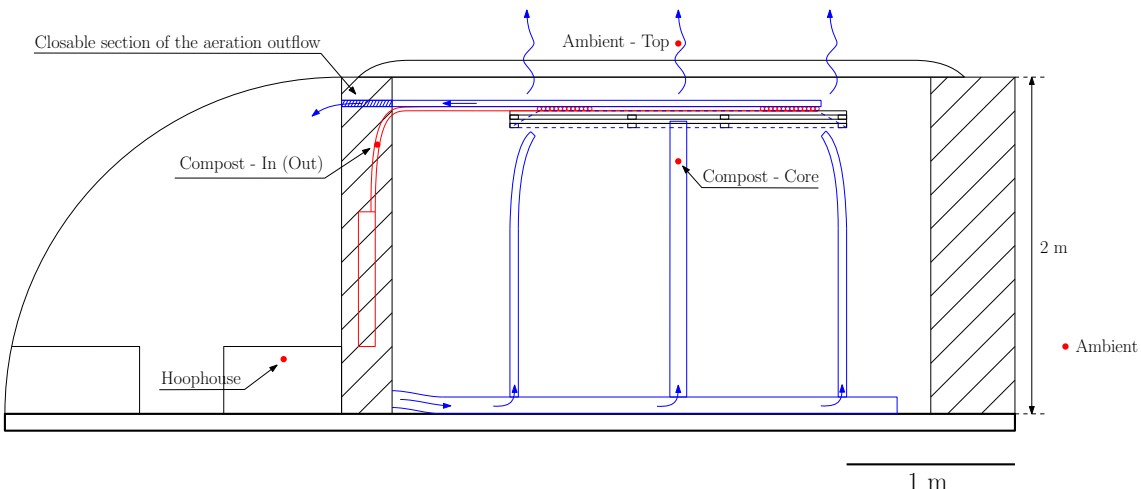

**Figure 7.** Side view of the composting bioreactor.

The southern side of the bioreactor is connected to the ad hoc constructed hoop house (orientation of the whole bioreactor is shown in Figure 6). This structure protects the CHRS (cf. Section 2.5) and can be used as a heated hoop house for winter crop production.

Compost mixture was chosen according to [17], i.e., main feedstock consists of woodchips, horse manure, fresh grass, leaves, matured compost. The resulting mixture has volumetric density of 603 kg/m$^3$ and moisture content 58% including added rainwater during the construction process. Calculation of dry matter content results in 4051 kg of dry matter (DM). Figure 8 shows design of internal composition of the bioreactor schematically layer by layer. Heat exchange area as well as aeration system is highlighted in the figure.

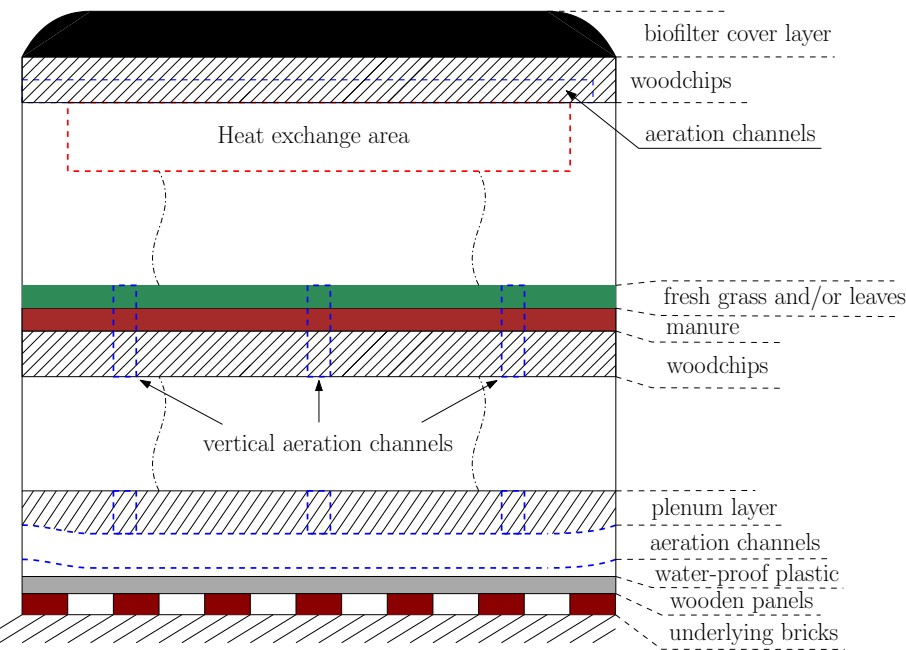

**Figure 8.** Feedstock layers (schematic drawing—disproportional). Woodchips, manure and fresh green grass and/or leaves are laid consecutively up to the desired height of the pile.

## 2.3. Construction of the Composting Bioreactor

For a standalone pile, the ground is leveled in order to secure drainage of compost leachate produced during watering of the pile. Moreover, thermal insulation from the ground is realized via gap created by bricks laid directly on the leveled ground and wooden panels as shown in Figure 9.

Polyethylene plastic foil is then laid onto wooden panels and fastened with first layer of strawbales along the perimeter of the pile. Perforated polyvinylchloride (PVC) pipes (diameter 10 cm) are used as a horizontal aeration channels spaced 1 m apart. These tubes are then covered with a plenum layer in a form of coarse woodchips allowing air to pass through to the upper parts of the composting pile. Figure 10 shows most of the floor layers in construction (yellow aeration tubes were straightened afterwards) and the first layer of manure and leaves.

The main body of the composting bioreactor consists of woodchips thoroughly mixed with horse manure and fresh green grass and/or wet leaves. Each 10 cm of material was soaked with water applied from a manual sprinkler. The whole composting pile then absorbed 3 m$^3$ of rainwater in order to achieve desirable moisture level of the composting mixture.

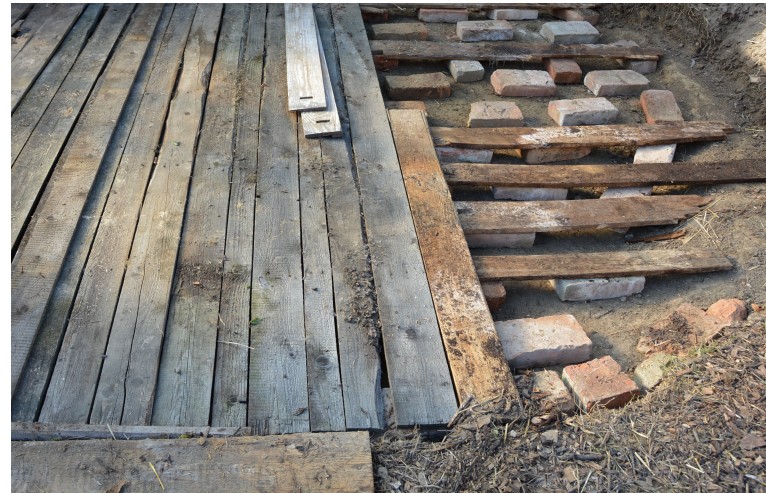

**Figure 9.** Compost floor—underlaying bricks and wooden panels provide thermal insulation of the composting material from the ground. Figure shows the construction in progress that is almost ready for installation of waterproof plastic layer.

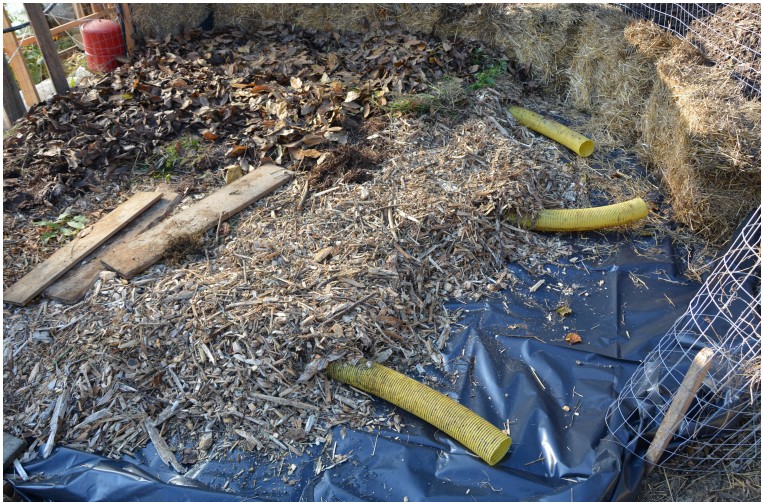

**Figure 10.** Waterproof plastic layer and perforated tubes covered with the aeration plenum layer (woodchips).

### 2.4. Compost Aeration and Natural Convection

To enhance and intensify the natural convection inside the pile [21], perforated PVC tubes (diameter 5 cm) are installed in vertical/slight diagonal position [22] as shown in Figure 11. The yellow pipe marks the center of the pile, whereas grey tubes are spaced in a square grid with a side length of 1 m. Half-length of the grey tubes was chosen in order to avoid deformation and potential damage of the heat exchanger (that will sit on top, see Figures 7 or 8) due to the pile compression in time.

The composting mix was laid up to the end of grey tubes and the aeration in the upper half of the pile (vertical direction up to the height of 2 m) was enhanced using flexible PVC pipes and short wooden sticks and poles that should bend easily under pressure. Once the desired height of the pile is achieved, the heat exchanger is laid on top and covered with permeable layer as described in Section 2.1.

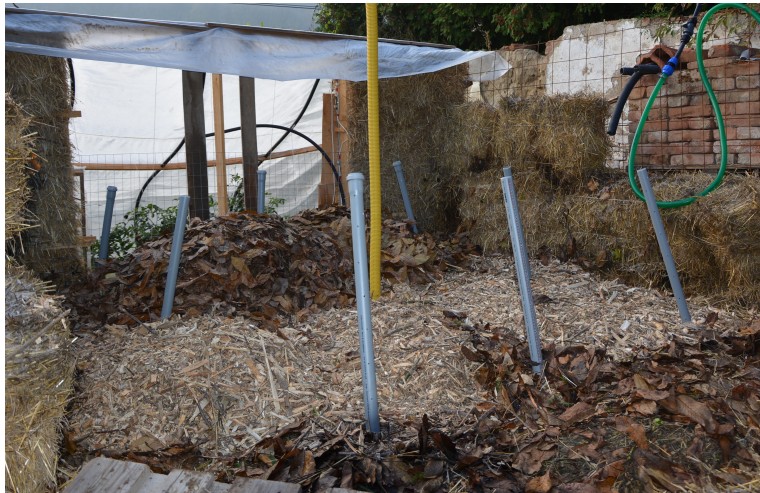

**Figure 11.** Perforated PVC tubes for enhancing the compost aeration in a vertical direction.

The biofilter cover layer in a form of unscreened compost then finishes the whole structure as shown in Figure 12. This cover layer serves primarily as thermal insulation and moisture/nutrients retention but also as a filter for odourous gases, methane and volatile organic compounds.

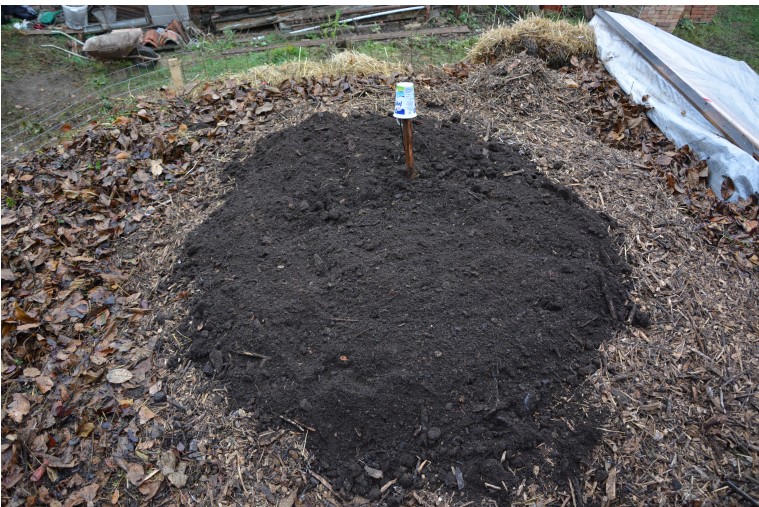

**Figure 12.** A compost biofilter cover layer of about 20–30 cm is applied on top of the permeable layer covering the heat exchanger.

Once the biofilter layer is spread on the whole surface of the pile and the strawbale thermal insulation is completed, the temperature measurement and heat recovery system can be connected. Figure 13 shows the bioreactor in its final stage of construction before the heating circuit is turned on.

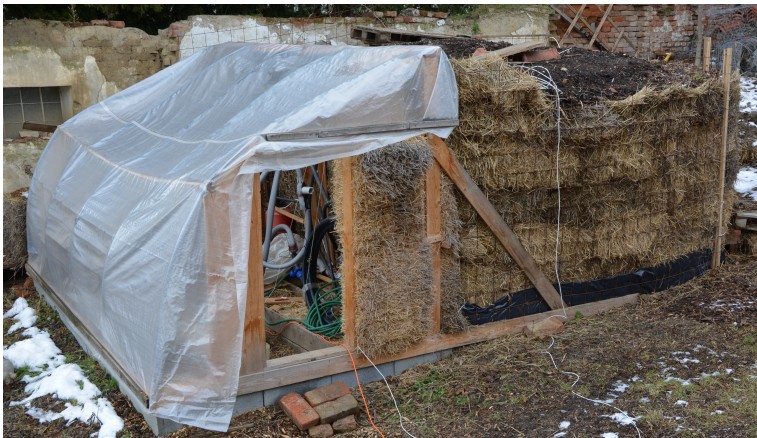

**Figure 13.** Composting bioreactor before final connection of heat recovery and temperature measurement circuits.

*2.5. Heating and Temperature Measurement System*

The CHRS contains the condenser-type heat exchanger, circulation pump, expansion vessel, ball valves, flow and fluid meter, mechanical temperature and pressure meter, radiating spiral and probes measuring fluid temperature leaving and entering the bioreactor (cf. Figure 14).

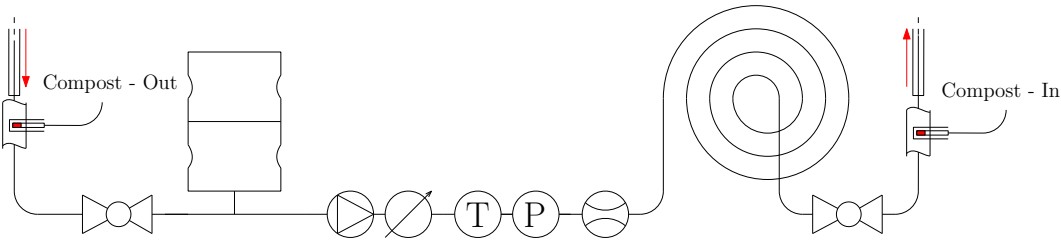

**Figure 14.** Design of the heat circuit.

Water is used as a heat-carrying medium in this experiment. As heated water leaves the heat exchanger, its temperature is measured in a thermowell (Compost—Out). The water then passes through the system, cools down and transfers the heat into its surrounding equipment and air cavity covered by the hoop house. Before the water is pumped back into the exchanger, its temperature is measured again (Compost—In).

Thermal digital thermometers (Dallas D18S20, accuracy $\pm 0.5\,^{\circ}\text{C}$ from $-10\,^{\circ}\text{C}$ to $+85\,^{\circ}\text{C}$) were used for temperature readings. Their labels and positions are summarized in Table 1:

**Table 1.** Summary of thermometer labels, locations and appearance in figures.

| Number | Name | Position | Figure |
|:---:|:---:|:---:|:---:|
| 1. | Compost - In | Inside hoop house, water temp., shaded and insulated. | 6, 7, 14 |
| 2. | Compost - Out | Inside hoop house, water temp., shaded and insulated. | 6, 7, 14 |
| 3. | Compost - Core | Inside the central aeration tube, inside the bioreactor. | 2, 6, 7, 15 |
| 4. | hoop house | Inside hoop house, shaded spot. | 6, 7, 15 |
| 5. | Ambient - Top | Outside, middle of the pile, directly on top, shaded spot. | 6, 7, 16, 17 |
| 6. | Ambient | Outside, distant and protected spot (against sun, wind). | 6, 7, 15 |

These sensors are designed for continuous temperature measurement and communication with the programmable logic controller (PLC) using a 1-Wire system. First, a 1-Wire line (Bus A) collecting data from sensors numbers 3, 6 and 7 belongs to the bioreactor itself and operates independently. Second, an independent 1-Wire line (Bus B) collects data from sensors inside the hoop house (see white

cables in Figure 13). Each sensor provides feedback every 60 s. The PLC unit can be either connected directly to a laptop or via WiFi to a remote server. For data transfer, collection and processing software routines, an in-house code is used.

## 3. Results

The bioreactor with proposed condenser-type heat exhanger was built during November 2018 (South Moravian region—Czech Republic) with already cold or freezing temperatures over night. In spite of these weather conditions, the initial heat-up phase with exponential temperature growth finished by the end of day 4. Once the temperature started to rise, the natural convection began to push the moist air through the aeration system. Although the biofilter was used, the moisture was transported into the hoop house, condensed and created icicles. For this reason, we closed the outflow orifice (see Figure 7) to keep the moisture inside the pile and let the aeration be driven only by diffusion.

The temperature dynamics during days 4 and 45 are depicted in Figure 15.

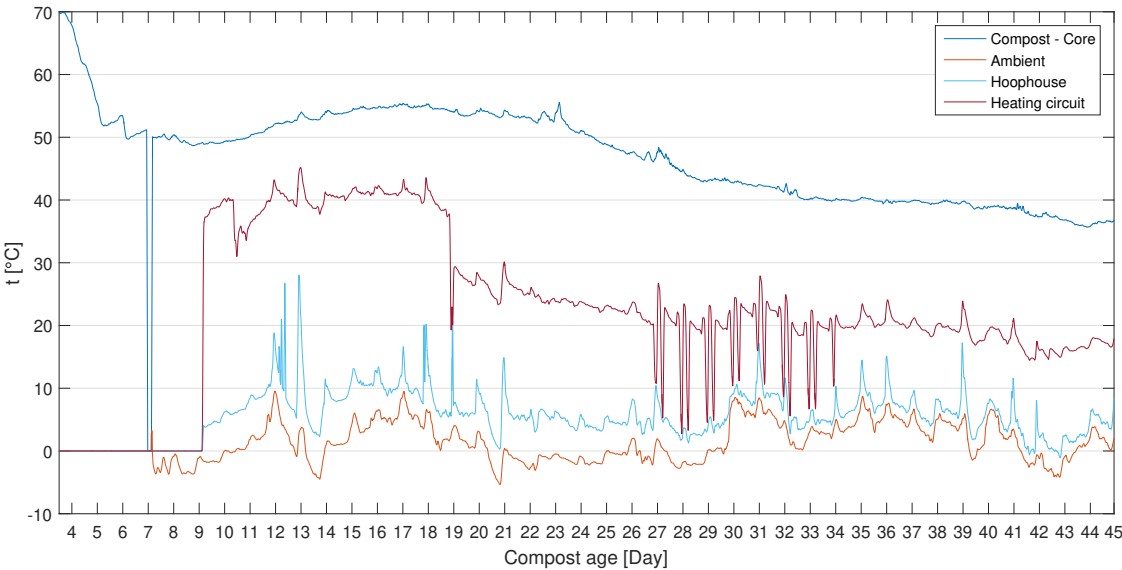

**Figure 15.** Core temperature of the pile, hoop house and the heating circuit compared to ambient during days 4 and 45. Note that the temperature drop during day 7 indicates sensor or connection failure on 1-wire line.

For the sake of lucidity, all data in Figure 15 are averaged over each hour. Measurement of the core temp. started by the end of day 3, the ambient temp. was launched later during day 7 (Bus A), whereas the rest of the sensors (Bus B) associated with the heating circuit and hoop house started during day 9 when the core temperature stabilized around 50 °C. The dark red line shows the temp. of the heating circuit, which is calculated as an average of Compost—In and Out temp. readings.

Three modes of circulation in the heating circuit were chosen: firstly, the circulation pump was on for 30 min in each hour, which resulted in fluctuating behaviour of readings from sensors Compost—In/Out. Detailed readings over days 9–10 are depicted in Figure 16.

The graph in Figure 16 also shows the initial/transient phase when the average temperature of the heat-carrying fluid is continuously rising with each cycle as the whole heating system becomes warmer. The green line (Ambient-Top) appears in the following figures as the indication of heat losses to the surroundings from the upper surface of the pile. The average temperature difference of around 5 °C between sensors Ambient and Ambient-top still indicates considerable heat loss, even if the biofilter cover layer is applied.

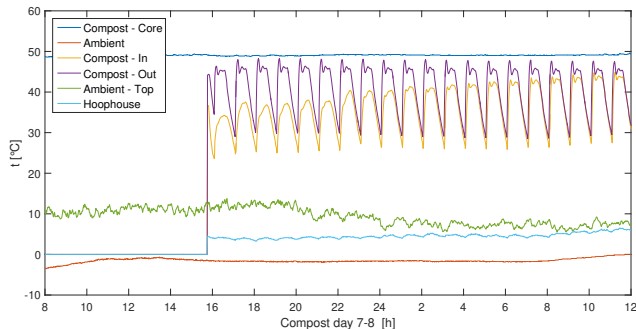

**Figure 16.** Detail of temperature profiles over days 9–10.

The most stabilized performance of the bioreactor over a 5-day period is shown in Figure 17.

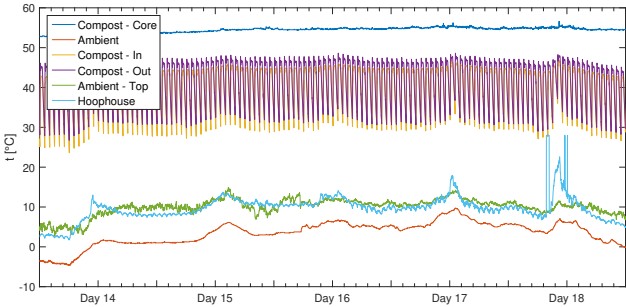

**Figure 17.** Temperature profiles of a 5-day period of active heat recovery (days 14–18).

For each half-day (0–12 h and 12–24 h), temperature differences were calculated and shown in the box plot in Figure 18.

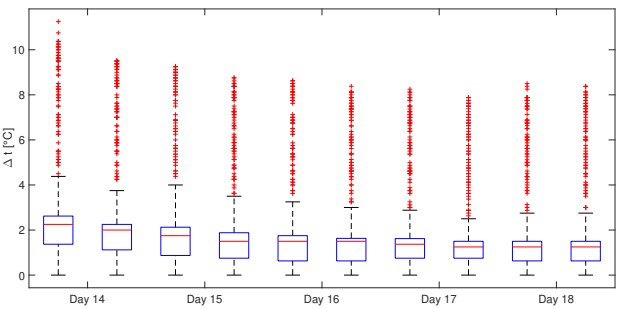

**Figure 18.** Box plot—measured temperature difference between Compost—Out and Compost—In.

Average volumetric flow rate measured using water meter for this period reads 5.029 l/min. The power output then follows from:

$$P = \dot{m}c_p\Delta T, \quad [W], \tag{1}$$

where $\dot{m}$ [kg/s] is the mass flow rate, $c_p$ [J/kgK] the specific heat capacity of water at constant pressure and $\Delta T$ [K] the temperature difference.

Bar plot in Figure 19 shows the thermal power output and the cumulative heat recovery [kWh] of the bioreactor over the selected 5-day period:

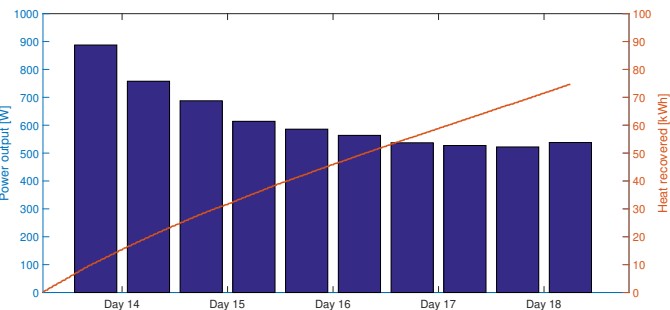

**Figure 19.** Bar plot—power output and heat recovery over the 5-day period.

During day 19, a sudden drop in the heating circuit appeared and the temperature stayed between 20 and 30 °C for several days, even though the core temp. was above 40 °C (see Figure 15).

In the second mode, the circulation pump was on for three hours, 2-times within the first 12 h in a day and the rest still as in the previous mode, i.e., the overall running time of the pump remains 12 h per day (see days 27–33 in Figure 15). However, the cooling period was found to be too long and, since day 34, the pump had run continuously (3rd mode) in order to stay above the freezing point. The temperature difference between Compost—In and Out stayed around 0.5 °C, which limited the heat recovery.

## 4. Discussion

The design ideas for a condenser-type heat exchanger for utilization of compost heat have been presented, cf. Figure 2. A prototype was built from LDPE pipe (see Figure 4) and installed as a key part of CHRS for the pilot-scale passively aerated standalone composting pile (modified Jean Pain mound). Its properties have been studied in a practical composting experiment over the winter period and its thermal performance has been evaluated.

This heat exchanger should overcome potential effects on microbial activity and heat production due to its position on the top of the pile in a "heat exchange area" as shown in Figure 8. Direct contact with actively composting biomass is avoided as the pipes lay on a platform that forms a cavity for hot gas accumulation (see Figure 3). As the moist air condenses on the heat-exchange surface, the heat is then transferred via conduction through the wall of the pipe to heat-carrying medium. Therefore, the exchanger utilizes the majority of "waste" heat available in the latent form.

However, if the heat exchanger surface is covered with other layers (woodchips, biochar and matured compost) for thermal insulation from the top, approximately 50% of the surface is therefore excluded from direct condensation. Better design of the holding platform with more spacious cavity should be adopted for maximal thermal performance of the heat exchanger itself.

Condensed water solutions are re-used directly inside the pile (thus local water cycle is achieved)—less moisture is diverted from the pile, thus keeping moisture content at a similar level. This is true if the outflow orifice of the aeration system is closed (see Figure 7). The spiral shape of the heat exchanger with horizontal gaps between each pipe allows gases to penetrate through this layer. Therefore, the natural convection should not be affected by the presence of the spiral.

From a practical point of view, the installation and dismantling of the heat exchanger is not dependent on the construction of the pile itself, thus allowing the extension of this technology for larger scales composting operations. It was found that the heat exchanger from LDPE used in this experiment is sturdy enough to withstand the pressure from the compaction of the medium-size pile in time and manual handling.

A limitation of this experiment is the radiating coil as heat appliance, which could not transfer enough heat to the hoop house air. This severely limited the measured power output, since the temperature difference between sensors Compost—In and Compost—Out stayed only around 2 °C on average (see Figure 18).

*Thermal Performace of the CHRS*

Figure 15 shows the period of 36 days (days 9–45) when the temperature measurements could be used for the estimation of the heat recovery directly from the operational data. It was found the average power output of 655 W with overall 574 kWh or 510 kJ/kg DM of heat recovered over this period. Note that this time period is just a part of the compost lifetime, when heat could be captured.

The readings for a 5-day period (days 14–18) were presented in Figure 17. Analysis of temperature differences (Figure 18) and the heat recovered in each day were coupled with cumulative heat recovery (Figure 19). It was found the average power output of 622 W and total 75 kWh or 66.7 kJ/kg DM of heat (over 5-day period) is recovered. Note that the core temperature measured inside the central aeration channel was not affected by the heat removal.

Power for the circulation pump: input 15 W or 0.18 kWh/day of electricity was the only input in order to run the CHRS (since the pump was on only 12 h/day). Average net energy recovery of the CHRS can therefore be calculated as 15.8 kWh/day in the form of heat. Extrapolating the average heat production from the beginning of the thermophilic phase (day 3) to day 45, we can estimate the average net heat recovery of 662.1 kWh per 42 days.

To evaluate the overall performance of presented CHRS, the heat source (bioreactor) provided considerable amount of energy that could be captured. Unfortunately, the utilization system (the radiating coil as heat appliance) was able to transfer only 1 kW in average. Therefore, heat recovery was limited especially during the first weeks of thermophilic phase, when the potential power output peaked around 4.5 kW. Thus, the presented results are severely limited by the underestimated power output of the appliance and not the heat source.

Comparing the results with review articles on various CHRS's [5], the lab-scale reactors reached a heat recovery rate 1895 kJ/h = 0.526 kW and pilot-scale reactors reached 20,035 kJ/h = 5.565 kW. Our composting system with a condenser-type heat exchanger had peak power output of 13,940 kJ/h = 4.550 W and longterm average power output of 2358 kJ/h = 655 W. With proper heat appliance, we estimate that the measured power output would increase four times. Future work will be therefore focusing on improvements of the heat appliance as well as the heat exchanger design and manufacturing for higher recovery rates.

The plastic foil stretched over the southern side of the bioreactor (see Figure 13) protected not only the heat utilization system, but it also created a hoop house structure. The bioreactor itself provided thermal mass and insulation from the northern side. Heat could be therefore dissipated through the uninsulated north wall and contributed to the elevated temperature inside the hoop house. Temperature readings shown in Figure 15 indicate only two days (day 42, 43) over the winter months (December and January) with below-zero temperatures.

It is important to note that the position of the probe inside the hoop house is approximately 50 cm from the ground and close to the reactor. More experimental research is needed for verification, if this kind of structure is able to keep the air (and soil) in the hoop house warm throughout the whole winter. Applications of compost-generated heat in greenhouses and hoop houses are promising, especially due to its synergistic relationship: compost produces heat, nutrients and CO2 and the enhanced growth of plants then provides feedstock for composting.With the results obtained from this study, we plan to explore concepts for heating of green and hoop houses using compost heat.

Lastly, the performance of composting reactors depends on various factors including size and shape of the pile, initial compost mixture, moisture, porosity of the material, etc. Regarding only the volume of the pile, cylindrical piles with larger diameter and height perform better due to their lower area-to-volume ratio. Heat losses are therefore reduced and more heat can be used for recovery. Larger piles are also able to produce heat for longer periods (piles 6 m in diameter, 2.5 m high should provide usable heat for more than 12 months, [19]).

## 5. Conclusions

This paper introduces a new design of condenser-type heat exchanger for compost heat recovery systems. The main advantage compared to the buried-types hydronic heat exchangers is the extraction of a more abundant latent part from the overall enthalpy contained in compost vapours, thus enhancing the heat recovery rate. The exchanger uses natural convection (chimney effect) inside the compost pile. Hot and moist gases rise through vertical aeration channels to the cavity created on top of the mound and condense on the heat exchanger surface. Due to this heat removal method, a potential adverse effect on heat generation (driven by microbial activity) is reduced, since no direct heat removal in active composting zone is applied.

Moreover, moisture retention is achieved since the condensed water stays in the pile. From the practical point of view, the major advantage of this exchanger is that it can be installed after the compost mound is finished. This design simplifies the construction and allows machinery manipulation without damaging the exchanger.

The exchanger was tested with a passively aerated static pile (modified Jean Pain mound) and hydronic heat utilization system inside the ad hoc constructed hoop house. Temperature dynamics and performance were measured and evaluated in order to compare with other CHRS's [5]. Unfortunately, power output of the final appliance was chosen below the possible power output of the heat source (bioreactor). The heat recovery was therefore severely limited by the transfer and utilization parts of the CHRS, not by the heat source itself. As a recommendation for the following experiments, these parts of the system need to be adjusted in order to gain maximal heat recovery rate from the bioreactor.

**Author Contributions:** Individual contributions of authors are listed as follows: Conceptualization, J.B. and J.F.; Data curation, J.F.; Investigation, J.B.; Methodology, J.B. and J.F.; Project administration, M.J.; Resources, J.B.; Software, J.B.; Supervision, M.J.; Validation, J.B., J.F. and M.J.; Visualization, J.B.; Writing—original draft, J.B.; Writing—review and editing, J.F. and M.J.

**Funding:** This research was funded by the Brno University of Technology project FSI-S-17-4444.

**Conflicts of Interest:** The authors declare no conflict of interest. The funders had no role in the design of the study; in the collection, analyses, or interpretation of data; in the writing of the manuscript, or in the decision to publish the results.

## Abbreviations

The following abbreviations are used in this manuscript:

CHRS　　Compost Heat Recovery System
CHE　　　Condenser-type Heat Exchanger
LDPE　　Low Density Polyethylene
PVC　　　Polyvinylchloride
PLC　　　Programmable logic controller

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
