# Peer review of "Condenser-Type Heat Exchanger for Compost Heat Recovery Systems"

_energies, doi:10.3390/en12081583_

Round 1
Reviewer 1 Report
The carried out investigation on the particular heat exchanger is interesting and the paper is in good shape. The authors could only add a comment on the expected performance on a large scale and long period application.
Author Response
Response:
Performance of composting reactors depends on various factors including size and shape of the pile, initial compost mixture, moisture, porosity of the material, etc. Regarding only the volume of the pile, cylindrical piles with larger diameter and height perform better due to its lower area-to-volume ratio. Heat losses are therefore reduced and more heat can be used for recovery. Larger piles are able to produce heat for longer periods (piles 6 m in diameter, 2.5 m high should provide usable heat for more than 12 months). The technique of compost heat capture is important since excess heat removal can affect the composting process. In large scale applications, the design and manufacturing of condenser-type heat exchanger need to be adjusted. For large batches on commercial scale, negative aeration with vapor-condensing heat exchanger is recommended. Power output of these facilities should reach more than 50 kW in longterm average.
The comment on the expected performance on a large scale and long period application was added to discussion section.

Reviewer 2 Report
Authors provided a very interesting idea of the compost heat exchanger. The topic is relevant to the Journal topic. There are a lot of experimental data and concept tests but the scientific side should be emphasised prior to the final publication.
The recommendation is a major review on the ground presented below.
1. In the introduction, you need to connect the state of the art to your paper goals. Please follow the literature review by a clear and concise state of the art analysis. This should clearly show the knowledge gaps identified and link them to your paper goals. Please reason both the novelty and the relevance of your paper goals. The current write-up does not highlight the limitations and research gap that this paper wants to fill.
2. Scientific novelty and contribution are not clearly defined. Please provide the difference from previous research published before.
3. Please avoid all reference lump (for example, 1,2; 14-17; 18-20 etc.). It means that the state of the art is not appropriate. Describe all references separately.
4. Please do not use "Construction" for heat exchangers. "Design and manufacturing" is recommended.
5. In the discussion section please provide the limitations, future work and comparison the results with other research.
Author Response
Response to Reviewer 2 comments:
Comments:
Authors provided a very interesting idea of the compost heat exchanger. The topic is relevant to the Journal topic. There are a lot of experimental data and concept tests but the scientific side should be emphasised prior to the final publication.
The recommendation is a major review on the ground presented below.
Point 1. In the introduction, you need to connect the state of the art to your paper goals. Please follow the literature review by a clear and concise state of the art analysis. This should clearly show the knowledge gaps identified and link them to your paper goals. Please reason both the novelty and the relevance of your paper goals. The current write-up does not highlight the limitations and research gap that this paper wants to fill.
Response to Point 1:
The section Introduction (i.e. literature review and state of the art) was reformulated, a few new references added and regrouped.
Following goals of the study were clearly stated:
i. Introduction of a new type of heat exhanger based on condensation of water vapour.
ii. Provide evidence, operational data and analysis of the performance of the heat exchanger prototype under real conditions (in a medium-sized compost pile).
iii. Share and discuss observations and draw conclusions for the further development of compost heat recovery technique.
Current scientific research in the field of compost heating lacks the evidence and clear detailed data showing how much heat can be recovered, how the power output changes in time and according to phases of composting. The objective of this study is also the publication of clear dataset for comparison with other research.
Point 2. Scientific novelty and contribution are not clearly defined. Please provide the difference from previous research published before.
Response to Point 2:
Previous research on Jean Pain style composting was performed with burried-type heat exchangers. By using the original heat exchanger, conduction as heat transport mechanism prevails. Limitations of this system are the susceptibility to leakage and potential damage by heavy weight or manipulation. The process of instalation/dismantle of the heat exchanger is dependent on the pile construction/deconstruction and it can be labour intensive and impractical in larger scales (when machinery is used). Importantly, excess heat removal from internal parts of the pile can affect heat generation and potentialy the microbial activity itself.
This study shows novel approach to heat capture from Jean Pain style compost system. It provides design of a condenser-type heat exchanger and adjustments of the compost mound. This system is designed to overcome limitations mentioned above. Concepts, methodology and experimental data are provided which allows replication of the experiment, comparison and further improvements.
Point 3. Please avoid all reference lump (for example, 1,2; 14-17; 18-20 etc.). It means that the state of the art is not appropriate. Describe all references separately.
Response to Point 3:
Sentences with reference lumps were revised. If the lump is still used, we either list several publications on the same topic (cf. [15-18]) or refer to publications discussing the same idea and providing more evidence (cf. [12,19]).
Point 4. Please do not use "Construction" for heat exchangers. "Design and manufacturing" is recommended.
Response to Point 4:
For heat exchangers, the term „Design and manufacturing“ is now used instead „Construction“.
Point 5. In the discussion section please provide the limitations, future work and comparison the results with other research.
Response to Point 5:
The discussion section was revised and limitations, future work and comparison with other research was added.
Limitation of this particular experiment is the radiating coil as heat appliance, which could not transfer enough heat to the hoop house air. This severly limited the measured power output, since the temperature difference between sensors Compost-In and Compost-Out was only around 2°C in average.
Comparing the results with review article on various CHRS‘s [5], the lab-scale reactors reached heat recovery rate 1895 kJ/h and pilot-scale reactors reached 20035 kJ/h. Our composting system with condenser-type heat exchanger had peak power output of 13940 kJ/h and longterm average power output of 2358 kJ/h. However, with proper heat appliance, we estimate that the measured power output would increase 4-times.
Future work will be therefore focusing on improvements of the heat appliance as well as the heat exchanger design and manufacturing for higher recovery rates. Also, new concepts for heating of green and hoop houses using compost heat will be tested.

Reviewer 3 Report
The study details the design of condenser-type heat exchanger for compost heat recovery 316 systems. A significant portion of the article describes the design itself with the limited constructive outcome from the heat exchanger and limited application. Here are a few things needed to be more appropriately covered.
The objective of this study is not defined.
In Figure 2, the unit of length should be similar
In Figure 15, what causes the sudden core temperature drop at day 7?
In Figure 16, why the compost-in and compost-out temperature fluctuates too much?
Author Response
Response to Reviewer 3 comments:
Comments:
The study details the design of condenser-type heat exchanger for compost heat recovery 316 systems. A significant portion of the article describes the design itself with the limited constructive outcome from the heat exchanger and limited application. Here are a few things needed to be more appropriately covered.
Point 1. The objective of this study is not defined.
Response to Point 1:
The section Introduction (i.e. literature review and state of the art) was reformulated and the following goals of the study were clearly stated:
i. Introduction of a new type of heat exhanger based on condensation of water vapour.
ii. Provide evidence, operational data and analysis of the performance of the heat exchanger prototype under real conditions (in a medium-sized compost pile).
iii. Share and discuss observations and draw conclusions for the further development of compost heat recovery technique.
Also, the discussion section was revised and limitations, future work and comparison with other research was added.
Point 2. In Figure 2, the unit of length should be similar
Response to Point 2:
The units of length were adjusted to mm in Fig. 2.
Point 3. In Figure 15, what causes the sudden core temperature drop at day 7?
Response to Point 3:
The temperature drop of this type (to zero values and staying constant for a while) indicates some sensor or connection failure on 1-wire line. A note was added to the caption of the figure.
Point 4. In Figure 16, why the compost-in and compost-out temperature fluctuates too much?
Response to Point 4:
The fluctuations of Compost-In and Compost-Out values are caused by the mode of water circulation. In Fig. 16, the first mode is shown - the circulation pump is on for 30 min and off for another 30 min.

Round 2
Reviewer 2 Report
The authors improved the manuscript according to reviewer comments. Now it is acceptable for the publication.
Reviewer 3 Report
The change looks good.